# Beneficial Actions of 4-Methylumbelliferone in Type 1 Diabetes by Promoting β Cell Renewal and Inhibiting Dedifferentiation

**DOI:** 10.3390/biomedicines12122790

**Published:** 2024-12-09

**Authors:** Wencheng Zhang, Shuo Yang, Xinwen Yu, Shanshan Zhu, Xin Wang, Fei Sun, Shengru Liang, Xiaoguang Wang, Guohong Zhao, Bin Gao

**Affiliations:** Department of Endocrinology, Second Affiliated Hospital of Air Force Military Medical University, Xi’an 710038, China; wencheng1437@163.com (W.Z.); yangshuo109@163.com (S.Y.); yuxinwen11@126.com (X.Y.); zss3013@stu.xjtu.edu.cn (S.Z.); 18049451310@163.com (X.W.); mustbesunfei@sina.cn (F.S.); lsandra131@126.com (S.L.); tdwxg628@163.com (X.W.)

**Keywords:** 4-methylumbelliferone, Type 1 diabetes, β cells, streptozotocin, dedifferentiation

## Abstract

**Background/Objectives**: This study aims to investigate the effects of 4-methylumbelliferone (4-MU) on islet morphology, cell phenotype and function, and to explore possible mechanisms of β cell regeneration. **Methods**: The Type 1 diabetes (T1D) model was induced by continuous dose injection of streptozotocin (STZ), and mice were treated with 4-MU for 3 weeks. Plasma insulin level, islet cell phenotype and immune infiltration were determined by IPGTT, ELISA, HE and immunofluorescence. The Ins2^Cre/+^/Rosa26-eGFP transgenic mice model was used to detect β identity change. Primary rodent islets were incubated with 4-MU or vehicle in the presence or absence of STZ, AO/PI staining, and a scanning electron microscope (SEM), PCR and ELISA were used to evaluated islet viability, islet morphology, the specific markers of islet β cells and insulin secretion. **Results**: Treatment with 4-MU significantly decreased blood glucose and increased plasma insulin levels in STZ-induced diabetes. The plasma insulin level in the STZ group was 7.211 ± 2.602 ng/mL, which was significantly lower than the control group level (26.94 ± 4.300 ng/mL, *p* < 0.001). In contrast, the plasma insulin level in the STZ + 4-MU group was 22.29 ± 7.791 ng/mL, which was significantly higher than the STZ group (*p* < 0.05). The 4-MU treatment increased islet and β cells numbers and decreased α cell numbers in STZ-induced diabetes. **Conclusions**: Islet inflammation as indicated by insulin and CD3 was caused by infiltrates, and the β cell proliferation as indicated by insulin and Ki67 was boosted by 4-MU. β cell dedifferentiation was inhibited by 4-MU as assessed by insulin and glucagon double-positive cells and confirmed by Ins2^Cre/+^/Rosa26-eGFP mice. In cultured primary rodent islets, 4-MU restored islet viability, protected islet morphology, inhibited β-cell dedifferentiation, and promoted insulin secretion. The benefits of 4-MU in T1D have been proved to be associated with β cells self-replication, dedifferentiation inhibition and immune progression suppression, which help to maintain β cell mass.

## 1. Introduction

Type 1 diabetes (T1D) is an autoimmune disease in which β cells are selectively destroyed, leading to chronic hyperglycemia and the need for exogenous insulin therapy. Despite advances in insulin delivery systems and glucose monitoring technologies, there remains a significant unmet need for therapies that can regenerate or protect β-cells and restore endogenous insulin production. The small number of β-cells preserved in the pancreas of patients with T1D provides pancreatic cues for immunization or β-cells regeneration in T1D [1,2]. Although streptozotocin (STZ)-induced diabetes cannot completely mimic T1D, it has been widely used to mimic pancreatic β cell destruction in T1D animals because of the similar β cell damage [3,4]. STZ-induced diabetes can be used to study how to restore islet function in T1D therapy, as it does not have the sustained immune attack of T1D [5]. It has been reported that residual β cells in STZ-induced diabetic mice can recover spontaneously [6,7], that STZ-induced diabetes can lead to β cells dedifferentiation, and that trans-differentiated β cells benefit islet regeneration [8]. Moreover, it has been demonstrated that HA deposition is both temporally and anatomically related to the progression of insulitis in a mouse model of T1D [9,10]. Current treatments for T1D primarily focus on managing blood glucose levels through insulin therapy and lifestyle modifications. However, these approaches do not address the root cause of the disease, which is the progressive loss of functional β-cells. Novel strategies aimed at promoting β-cell regeneration and preventing dedifferentiation are therefore essential for achieving a cure or long-term remission of T1D [11,12,13]. However, there is still an urgent need to find more effective and cost-effective treatments and drugs, taking into account factors such as safety, efficacy, and treatment cost.

As a natural coumarin derivative with no reported side effects, 4-methylumbelliferone (4-MU) is well-known as an inhibitor of HA synthesis and deposition [14,15,16]. It is also commercially available as “hymecromone”, and used as a choleretic and antispasmodic drug for treating biliary spasm in Europe and Asia [17]. Moreover, it has great influence on many pathologies, such as autoimmune diseases, metabolic diseases and even cancer [18]. Recently, 4-MU has been reported to promote the induction of regulatory T-cells and prevent autoimmunity, so restoring immune tolerance during autoimmune insulitis in T1D mice [9,10]. In addition, 4-MU can also increase brown fat capacity while improving the development of hyperglycemia and obesity [19]. However, the effects of 4-MU on islet protection and the effect on islet cell composition have not been studied.

Herein, the primary objective of this study is to investigate the beneficial actions of 4-MU in T1D by promoting β-cell renewal and inhibiting dedifferentiation. Ins2^Cre/+^/Rosa26-eGFP C57BL/6 was constructed as a classical islet β cell lineage tracer mouse, which can be used to systematically track insulin-expressing cells. Firstly, mice labeled with β cell-specific green fluorescent protein (eGFP) produced by the Cre-LoxP system were used to explore the improvement in β cell dedifferentiation. And then, the viability, morphology and function in STZ/treated primary islets were detected by loading or not loading 4-MU in vitro. Finally, the expression level of β cell-specific transcription factors were detected. Our findings may provide a foundation for the development of novel therapeutic strategies for the management and potential cure of T1D.

## 2. Materials and Methods

### 2.1. Animals

Six-week male C57BL/6J mice with body weight of 18–22 g was purchased from the Experimental Animal Center of Forth Military Medical University (Xi’an, China) and acclimated for one week prior to the experiment. Ins2^Cre/+^/Rosa26-eGFP C57BL/6 mice (Cyagen Biosciences, Guangzhou, China) were bred in house at the Experimental Animal Center of Forth Military Medical University (Xi’an, China, Approval NO. SYXK-2024-003). They were fed with standard laboratory chow and drank water. They were maintained under standard conditions (12 h light–dark cycle; 23–25 °C; 50–60% relative humidity). All experimental operations were approved by the Animal Care and Use Committee of Forth Military Medical University (Approval NO. IACUC-20230955).

### 2.2. Experimental Design

In the first batches of experiments, the mice were randomly divided into three experimental groups: (I) control group (CON group), (II) STZ-induced diabetic group (STZ group), and (III) STZ-induced diabetic group treated with 4-MU (STZ + 4-MU group). STZ was dissolved in cold citrate buffer (0.1 M, pH 4.5) immediately prior to injection. In the in vivo model, continuous intraperitoneal injection of streptozotocin (50 mg/kg) once a day for 5 days is a common scheme. A large number of studies have shown that STZ in this dosage range can effectively induce the characteristics of T1D, such as hyperglycemia [20,21]. Mice in the STZ group and the STZ + 4-MU group were intraperitoneally (i.p.) injected with 0.2 mL STZ (50 mg/kg, Sigma-Aldrich, St. Louis, MO, USA) for 5 consecutive days, and the mice in the CON group were intraperitoneally injected with 0.2 mL saline. One week later, the blood glucose concentration of these mice was determined by using blood samples collected from the tail vein after an 8 h fasting. The diabetic condition was confirmed when the fasting blood glucose was ≥11.1 mmol/L or the random blood glucose was ≥16.7 mmol/L at least twice [22]. The control and STZ groups were fed with regular chow, and the STZ + 4-MU group were fed with regular chow supplemented with 4-MU (250 mg/mouse/day, Alfa Aesar, Ward Hill, MA). We determined the intervention concentration of drugs according to relevant references [10,23]. After a continuous 3-week feeding, an intraperitoneal glucose tolerance test (IPGTT) was performed and samples were collected. Plasma insulin levels were detected by insulin secretion enzyme-linked immunosorbent assay (ELISA), and the expression and localization of insulin and glucagon in pancreatic tissues were detected by immunofluorescence.

In the second batches of experiments, 18 male C57BL/6J background Ins2^Cre/+^/Rosa26-eGFP islet β cells lineage tracing mice aged 6–8 weeks were constructed, randomly divided into three groups, and received the same treatment as described above. Immunofluorescence was used to detect the expression and localization of insulin, glucagon, NKX6.1, and eGFP in pancreatic tissues.

### 2.3. Measurement of Blood Glucose Levels

Blood samples were obtained from the tail vein of mice at the same time each week. Briefly, 1–2 mm was clipped from the tip of the mouse tail, the first drop was discarded, and the second drop was used to test blood glucose levels [24]. Non-fasting blood glucose concentration was measured using a glucometer (Roche Diagnostics, Barcelona, Spain).

### 2.4. Intraperitoneal Glucose Tolerance Test (IPGTT)

To perform the intraperitoneal glucose tolerance test (IPGTT), basal blood glucose levels were first detected after 12–16 h overnight fasting [25]. A 20% (wt/vol) of glucose solution was intraperitoneally administered at 2 g/kg, and blood glucose levels were determined at 15, 30, 60, 90 and 120 min after the glucose loading. Blood glucose level was determined using a One-Touch Profile portable blood glucose monitor (Roche Diagnostics, Inc.). The results were expressed as the area under the curve (AUC).

### 2.5. Histological Examination

Excised pancreatic tissues were fixed with 4% paraformaldehyde overnight, cut into 5 μm thick sections, and stained with 0.1% hematoxylin for 15 min and 0.5% eosin for 5 min, both at room temperature, according to standard procedures. The pathological and morphological evaluation was further carried out by microscope.

### 2.6. Immunofluorescent Staining and Quantification

For immunohistochemistry, slides were dewaxed by xylene and rehydrated through a series of ethanol solutions (100–50%). Heat-induced antigen retrieval was carried out in sodium citrate buffer. The sections were blocked in 4% bovine serum albumin solution and subsequently incubated with primary antibodies at 4 °C overnight. Following this, sections were rinsed in PBS and incubated for 1 h at room temperature with secondary antibodies. Slides were finally incubated with DAPI (1 μg/mL, Sigma-Aldrich) for 20 min at room temperature, and then mounted for imaging using NIKON Eclipse C1 confocal fluorescence microscope (Nikon, AX/AX R with NSPARC, Tokyo, Japan) fitted with DAPI (350 nm), FITC (488 nm) and TRITC (594 nm) filters.

The primary antibodies were listed as follows: mouse anti-insulin antibody (8138T, 1:400, Cell Signaling Technology, Danvers, MA, USA), rabbit anti-glucagon antibody (2760S, 1:200, Cell Signaling Technology), rabbit anti-NKX6.1 antibody (54551T, 1:400, Cell Signaling), and Ki67 (GB121499-100, 1:500, Servicebio, Wuhan, China). The secondary antibodies were listed as follows: Alexa Fluor 488 conjugated goat anti-rabbit IgG (1:400, Servicebio), CY3 conjugated goat anti-mouse IgG (1:300, Servicebio, Wuhan, China), CY3 conjugated goat anti-rabbit IgG (1:300, Servicebio, Wuhan, China), and CY5 conjugated goat anti-mouse IgG (1:400, Servicebio, Wuhan, China).

### 2.7. Isolation and Culture of Primary Pancreatic Islets

Mouse pancreatic islets were isolated using the Liberase digestion method [26]. Briefly, mice were anesthetized by intraperitoneal injection of sodium pentobarbital solution (100 mg/kg body weight) and the pancreas was perfused with 3 mL collagenase P (1 mg/mL, Sigma-Aldrich, USA). The islets were handpicked after purification with Histopaque 1077 (Sigma-Aldrich, USA) density gradients. The islets were isolated and washed twice with 5 mL HBSS buffer. These islets were then cultured overnight in RPMI 1640 containing 10% FBS, for further experiment.

### 2.8. Treatment of Islets with STZ and 4-MU

Isolated islets were cultured for another 24 h in 48-well plates with ~10 islets per well in RPMI 1640, supplemented with STZ (1 mM), with/without 4-MU (0.3 mM). We determined the intervention concentration of drugs according to the relevant references [27,28].

### 2.9. Islet Viability Assessment In Vitro

Pancreatic islets were separated into three groups (CON group, STZ group, and STZ + 4-MU group) and seeded into 12-well plates. To assess islet viability, AO/Pl staining (BestBio, Shanghai, China) was used to identify living and dead islet cells after 3-day culture with different formulations. Stained islets were observed on a fluorescence microscope (Eclipse 80i, Nikon, Tokyo, Japan) with emission wavelengths of 488 nm (live) and 594 nm (dead).

### 2.10. Insulin Secretion Enzyme-Linked Immunosorbent Assay (ELISA)

Mice plasma and cell culture supernatant were sampled for insulin detection. At the end of the experimental period, the mice were dislocated, and the whole blood (1.5–2 mL) was collected by eyeball removal in a centrifuge tube without anticoagulant. The collected blood samples were allowed to stand at room temperature for 1 h or overnight at 4 °C to allow serum precipitation. Afterwards, the samples were centrifuged at 4 °C, 1000× *g* for 20 min and the supernatant was collected for ELISA (Abcam, Cambridge, UK) according to the manufacturer’s instructions. Pancreatic islets were separated into three groups (CON group, STZ group, and STZ + 4-MU group) and seeded into 12-well plates. After 3 days’ culture with different treatment, the medium was changed to Krebs Ringer bicarbonate HEPES (KRBH) buffer for 1 h, followed by incubation in KRBH buffer containing low and high glucose (2.8 or 16.7 mM, respectively) for 30 min. Each treatment was run in triplicate for a single experiment. The insulin concentrations in the supernatant were measured using a Bovine Insulin ELISA kit. The islets were lysed in RIPA buffer and samples were kept at −80 °C. The islet insulin content was determined by normalizing the insulin content to the total protein content within the islets.

### 2.11. RNA Isolation, Reverse Transcription, and Quantitative PCR

Total RNA was extracted from islets using Minibest Universal RNA Extraction Kit (Takara, Toyoto, Japan), according to the manufacturer’s instructions. RNA quality and concentration were tested using SpectraMax Quick-Drop Micro-Volume Spectrophotometer (Molecular Devices, USA). Real-time PCR (RT-PCR) was performed using the CFX Connect™ Real-Time PCR Detection System (Bio-Rad Laboratories, Hercules, CA, USA). All experiments were performed in triplicate. Relative quantification for gene expression was calculated with the 2^−ΔΔCT^ method, which was normalized to the level of the housekeeping gene, β-actin [29]. The primer sequences used for real-time PCR were as listed follows:NKX6.1 Sense: ATCTTCTGGCCCGGAGTG;NKX6.1 Anti Sense: TCTCTCTGGTCCTGCCAAG;PDX1 Sense: GACCTTTCCCGAATGGAACC;PDX1 Anti Sense: GTTCCGCTGTGTAAGCACC;β-actin Sense: TGCGTGACATCAAAGAGAAG;β-actin Anti Sense: GATGCCACAGGATTCCATA.

## 3. Results

### 3.1. 4-MU Regulated Blood Glucose Levels and Improved Diabetic Symptoms

To investigate whether 4-MU administration could alleviate the T1D phenotype, we constructed a STZ-induced T1D mice model (Figure 1A). Firstly, we measured non-fasting blood glucose concentrations at the same time every week for three weeks. We observed that the glucose level in the STZ group was significantly elevated, and the glucose level was significantly decreased after 4-MU treatment (Figure 1B). And then, the IPGTT test results showed that both glucose and AUC were decreased in the 4-MU-treated group, demonstrating that 4-MU administration could enhance glucose tolerance in STZ mice (Figure 1C,D). In addition, we measured the plasma insulin level to assess the function of β-cells, and observed significantly reduced plasma insulin in the STZ-treated group. The plasma insulin level in the STZ group was 7.211 ± 2.602 ng/mL, which was significantly lower than the control group level (26.94 ± 4.300 ng/mL, *p* < 0.01). However, an obvious increased level in the 4-MU group (22.29 ± 7.791 ng/mL) was observed compared to in the STZ-treated group (*p* < 0.05), demonstrating that 4-MU treatment could prevent the STZ-induced reduction in plasma insulin (Figure 1E). Finally, the histologic analysis of pancreatic islets showed that mice in the 4-MU-treated group had a better islet quality, with less islet deformation and inflammatory infiltration, when compared with STZ-treated group (Figure 1F). These results indicated that the ameliorating effect of 4-MU on STZ-induced pancreatic β-cell damage was achieved by increasing insulin levels and improving blood glucose. 

### 3.2. 4-MU Increases β Cells Number, and Decreases α Cell Number in STZ-Induced T1D Mice

To assess islet morphology, we stained pancreatic islets from each treatment group (green indicates insulin; red indicates glucagon; blue indicates DAPI), and the representative images are shown in Figure 2A. We observed that treatment with 4-MU in STZ-induced T1D mice induced a significantly increased number of β cells (Figure 2B) and a significantly decreased number of α cells (Figure 2C), thereby producing an increased proportion of β cells to α cells (Figure 2D).

### 3.3. 4-MU Alleviates Inflammation and Promotes β Cell Renewal in STZ-Induced T1D Mice

In T1D, the islets were attacked by the autoimmune system, leading to β cell destruction, which in turn interferes with insulin secretion [30]. CD3^+^ T cells can be used to evaluate the degree of inflammation of islets and the infiltration of immune cells [31]. Therefore, we evaluated the immune infiltration levels in different groups by immunofluorescence. As shown in Figure 3A, pancreatic islets were infiltrated by more CD3^+^ T cells in the STZ group than in the CON group, and 4-MU treatment improved the immune infiltration. Moreover, we performed Ki67 staining to assess islet cell proliferation, and observed that 4-MU significantly increased β cell proliferation of STZ-induced diabetic mice (Figure 3B).

### 3.4. 4-MU Inhibited β Cell Dedifferentiation

In order to detect the effect of 4-MU on islet β cell dedifferentiation of islet β cells, β-cell lineage tracking mice (Ins2^Cre/+^/Rosa26-eGFP C57BL/6 mice) were constructed. Ins2-CreERT and R26-CAG-TdTomato hybrids are classical islet β cell lineages, tracing mice that can be used to trace insulin-expressing cells systematically [32]. Here, we used eGFP^+^/glucagon^+^ staining to suggest cell dedifferentiation. We observed fewer eGFP^+^/glucagon^+^ cells in the STZ group than in the CON group, but the situation was ameliorated in the 4-MU group (Figure 4A). Moreover, as a transcription factor to detect the development and function of pancreatic β cells, NKX6.1 is closely related to identity change [33]. In the CON group, NKX6.1 was evenly distributed in the nucleus, but in the STZ group, NKX6.1 appeared in the cytoplasm of β cells, suggesting that STZ could induce β cell dedifferentiation. However, it was found that 4-MU could effectively inhibit β cell dedifferentiation (Figure 4B), which was consistent with the eGFP^+^/glucagon^+^ results.

### 3.5. 4-MU Can Protect Islets from STZ-Induced Damage, Inhibit β Cell Dedifferentiation and Enhance Islet Insulin Secretion In Vitro

AO/PI staining has been widely used to detect cell viability, in which green fluorescence indicates living cells and red fluorescence indicates dead cells [34]. As shown in Figure 5A, compared with the CON group, the islet viability in the STZ group was significantly decreased, while 4-MU could significantly alleviate STZ-induced islet damage and restore islet viability in vitro.

In order to further observe the protective effect of 4-MU on islets, the ultrastructural changes in islets were observed by SEM (Figure 5B). We observed that the islets in the CON group had good structural stability and intercellular communication, and the contact area between islets and the surrounding environment was wide, which was favorable for material exchange and intercellular interaction. In contrast, in the STZ group, islets had a loose structure and a reduced contact area with the surrounding tissues, which greatly reduced the structural stability and intercellular communication of the islets, while 4-MU treatment protected the structural stability of the islets.

GSIS assay was also used to detect the insulin secretion level in vitro. Compared with the CON group, the glucose-stimulated insulin secretion level of pancreatic islets in the STZ group was significantly decreased, whereas the 4-MU group showed higher insulin release (Figure 5C).

In addition, we examined the effect of 4-MU on β-cell dedifferentiation in vitro. PDX1 and NKX6.1 play important roles in maintaining the characteristics and functions of islet β cells [35]. The RT-PCR results showed that 4-MU significantly up-regulated the expression of PDX1 and NKX6.1 transcription factors when compared with the STZ group, suggesting that 4-MU could inhibit STZ-induced β cell dedifferentiation (Figure 5D,E).

## 4. Discussion

Our study discovered that 4-MU improved blood glucose levels but also downregulated the α/β-cell ratio and inhibited β to α cell transdifferentiation in STZ-induced T1D mice. Additionally, 4-MU protected islet cells from STZ-induced damage, which not only maintains normal islet morphology, but also upregulates the expression of β cell-specific transcription factor and enhances islet insulin secretion in primary isolated mice islets. This study builds on previous findings by demonstrating 4-MU’s ameliorating effects on islets in a T1D mouse model, with a particular focus on its ability to inhibit β-cell dedifferentiation and promote self-replication.

T1D is an autoimmune disease in which T cells continuously attack and destroy insulin-secreting β-cells in the islets of Langerhans [36,37], and it is mainly characterized by insulitis. Many studies have found that there is a large amount of hyaluronan deposition in insulitis [38,39,40], and inhibiting the production of hyaluronan can re-establish immune tolerance during autoimmune insulitis [10,18,41]. Studies have demonstrated that 4-MU, an inhibitor of hyaluronan synthesis, can significantly reduce the deposition of hyaluronan in the islets of autoimmune diabetic mice, increase the insulin-positive area in the islets, and increase the number of regulatory T cells, thus having the potential to prevent the progression of diabetes [9,42]. In the diabetic rat model induced by high-fat diet and STZ, 4-MU can also significantly reduce blood glucose levels [43]. Therefore, a strategy based on 4-MU is regarded as an attracting anti-diabetic strategy. However, the antihyperglycemic mechanism of 4-MU is unclear, up to now.

In recent years, more and more studies have shown that T1D is not only triggered by the immune system attacking pancreatic β cells, but also that β cells dedifferentiate under chronic inflammation and metabolic stress, losing their normal insulin secretion capacity [44,45]. This process of dedifferentiation accelerates the progression of T1D [46]. Therefore, stopping β cells’ dedifferentiation and restoring their function is essential for T1D treatment. Therefore, whether 4-MU has a dedifferentiation effect on the hypoglycemic effect of T1D also deserves further investigation [47,48,49,50,51]. The mechanism by which 4-MU protects pancreatic islet β-cells from STZ damage is an interesting question, and the main purpose of our study. All current studies on 4-MU have attributed these protective effects to restoration of immune tolerance and increased expression of apoptosis inhibitors to ameliorate T1D loss [10,23,52], and few other mechanisms have been identified. Therefore, we investigated the effect of 4-MU on improving the changes in islet cell composition and dedifferentiation of β cells in STZ-induced diabetic mice, and found that 4-MU protected the number and function of β cells by promoting β cell proliferation and inhibiting β cell dedifferentiation. Thus, fate determination, mass maintenance, and functional protection of pancreatic β cells mass and function by 4-MU is a potential mechanism that provides a new strategy for T1D treatment.

Next, we carried out in vitro experiments to validate whether 4-MU can protect islet viability and inhibit β cell dedifferentiation. AO/PI staining is widely used to detect cell viability [34], and our study shows that 4-MU can restore the islet viability treated by STZ, stabilize the islet structure and restore islet function. In addition, 4-MU can up-regulate the mRNA levels of primary mouse islet β cell-specific markers NKX6.1 and PDX1, indicating that 4-MU may promote β cell maturation and inhibit dedifferentiation. We also carried out GSIS on primary mouse islets, and revealed that 4-MU increased insulin secretion. These results suggested that 4-MU may protect β cells from STZ damage mainly by inhibiting dedifferentiation.

At present, medication options for T1D, other than insulin, are still very limited, and the efficacy of an approved treatment scheme is not satisfactory, raising concerns about the prospect of T1D treatment. Although insulin is a central tool for glycemic management, it does not address the underlying etiology of T1D, which is the immune-mediated destruction and loss of function of pancreatic islet β-cells. Therefore, there is an urgent need to develop new therapies that can not only protect the β cells, but also promote their functional recovery. Previous studies have demonstrated that 4-MU can ameliorate Type 2 diabetes, suggesting that its use as a hyaluronic acid inhibitor not only alleviates HA deposition in t2d islets, but may also protect β cells from injury by increasing the expression of survivin, an inhibitor of apoptosis. Our latest study further showed that 4-MU also maintains the functional state of β cells by inhibiting their dedifferentiation and promoting β cell regeneration. This means that 4-MU has a dual role in not only protecting the integrity of β cells, but also promoting insulin secretion to help cope with the degradation of β cell function due to immune destruction and chronic stress. 4-MU is expected to be a key factor in effective long-term glycemic control, through maintaining the functional properties of β cells. This discovery brings new possibilities for the treatment of T1D, and future research and clinical validation will further clarify its application prospects.

However, there still exist several questions that require further investigation. Firstly, Gcg/α-lineage-labeled mice are necessary to validate whether α cells transdifferentiated into β cells after STZ and 4-MU treatment. Secondly, only NKX6.1 and PDX1 were preliminarily tested to reveal the possible mechanism of 4-MU inhibit dedifferentiation, and more transcription factors such as progenitor cell markers and α cell markers are required to confirm our findings. Thirdly, scRNA-seq analysis among STZ- and 4-MU-treatment mice islets are required to further understand the characteristics of dedifferentiation and transdifferentiation of islet cells. Lastly, it would be very meaningful to investigate the effects of 4-MU in human islets, especially from T1D patients.

## 5. Conclusions

In summary, this study demonstrates that the well-known compound 4-MU can be repurposed to improve hyperglycemia in experimental T1D. Our findings highlight the multifaceted benefits of 4-MU, including its ability to stimulate β-cell self-replication, inhibit β-cell dedifferentiation, and suppress immune-mediated damage. With its dual role in immune modulation and β-cell preservation, 4-MU shows promise for combination therapies in restoring β-cell function and blocking autoimmune processes in T1D, warranting further clinical research.

## Figures and Tables

**Figure 1 biomedicines-12-02790-f001:**
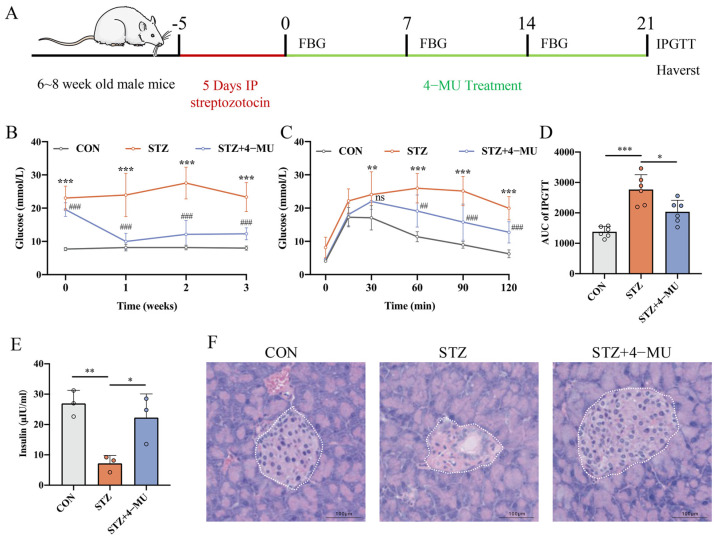
Diabetic phenotype and insulitis grading in STZ-induced T1D mice receiving 4-MU treatment. (**A**) Scheme of the experimental procedure. After 1 week of adaptive-type feeding, mice were injected with STZ at low doses for 5 consecutive days to construct the model of T1D, followed by treatment with 4-MU or saline for 3 weeks. The mice were executed after performing IPGTT experiments, and plasma was collected, and the pancreas was dissected, fixed and preserved. (**B**) Non-fasting blood glucose levels of mice in different groups. *n* = 6. (**C**) Blood glucose during an intraperitoneal glucose tolerance test (IPGTT) in STZ-induced T1D mice treated with 4-MU or saline for 3 weeks, and (**D**) area under the curve (AUC). *n* = 6 (**E**) Non-fasting plasma insulin levels of mice in different groups. *n* = 3. (**F**) Representative pictures of islet HE staining. Scale bars, 100 μm. Statistics were determined by two-way analysis of variance (ANOVA) with Tukey’s multiple comparisons test. * *p* < 0.05, ** *p* < 0.01, *** *p* < 0.001, ^##^
*p* < 0.01, ^###^
*p* < 0.001 vs. the STZ.

**Figure 2 biomedicines-12-02790-f002:**
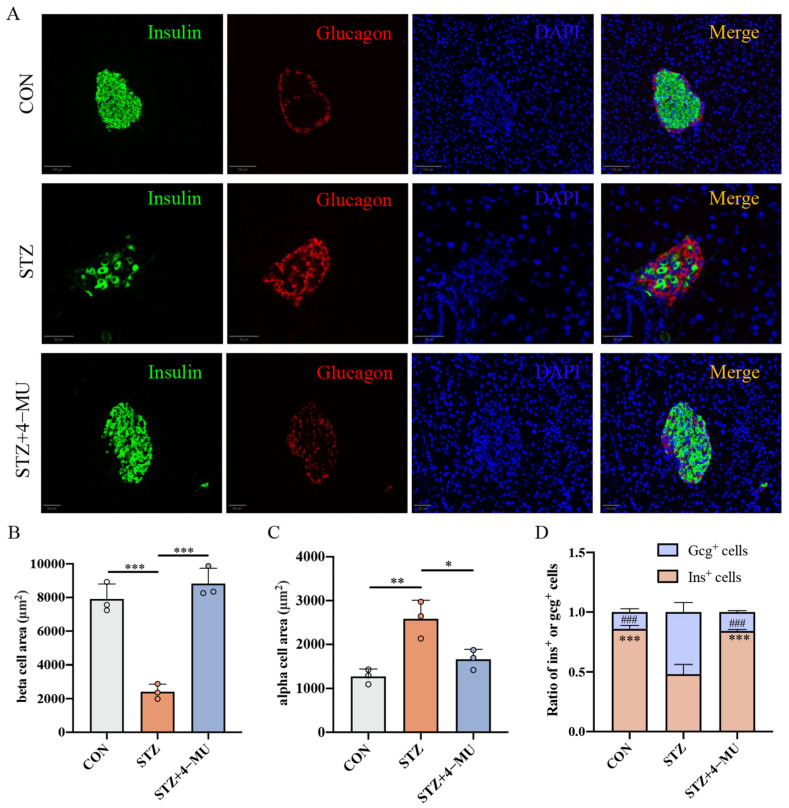
Histological analysis of insulin and glucagon markers in pancreatic tissue of STZ-induced T1D mice treated with 4-MU. (**A**) Representative islets of insulin (green) and glucagon (red) from different groups. Scale bars, 50 μm. Quantifications of (**B**) α cells, (**C**) β cells, and (**D**) the ratio of β cells (Ins^+^ cells) and α cells (Gcg^+^ cells). Statistics were determined by two-way analysis of variance (ANOVA) with Tukey’s multiple comparisons test. * *p* < 0.05, ** *p* < 0.01, *** *p* < 0.001, ^###^
*p* < 0.001 vs. the STZ.

**Figure 3 biomedicines-12-02790-f003:**
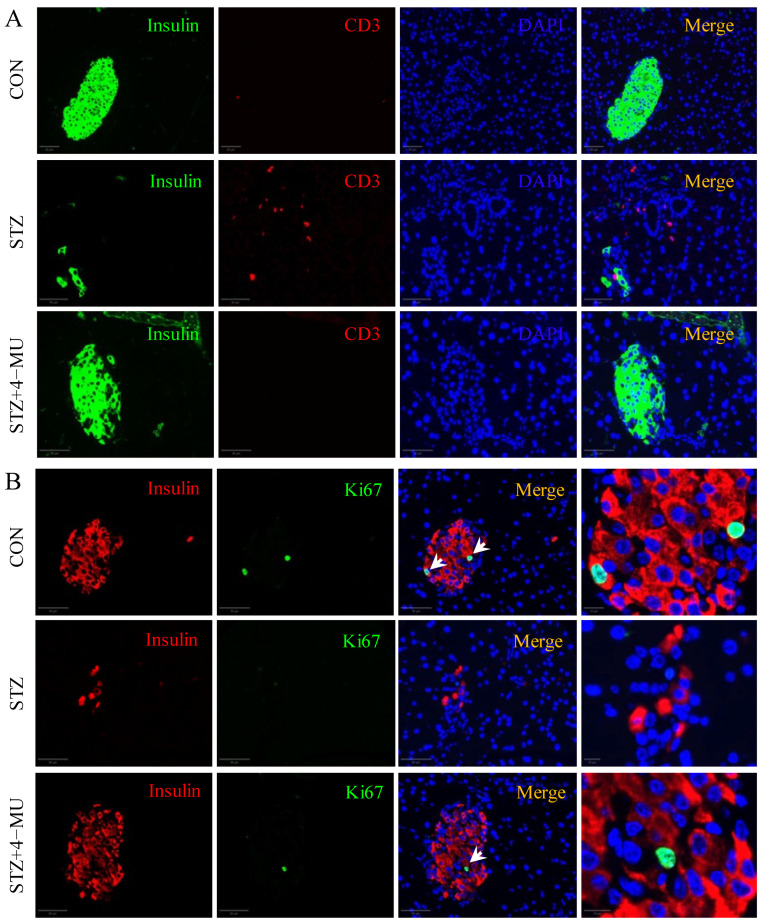
Effects of 4-MU therapy on immune cell infiltration and β cell proliferation of pancreatic islets in STZ-induced diabetic mice. (**A**) Parametric immune cell infiltration (insulin: green; CD3^+^ T cells: red) and (**B**) β cell proliferation (insulin: red; Ki67: green) were assessed in different group mice after 3 weeks of treatment with saline or 4-MU. Arrows indicate insulin^+^/Ki67^+^ cells. Scale bars, 50 μm, 10 μm.

**Figure 4 biomedicines-12-02790-f004:**
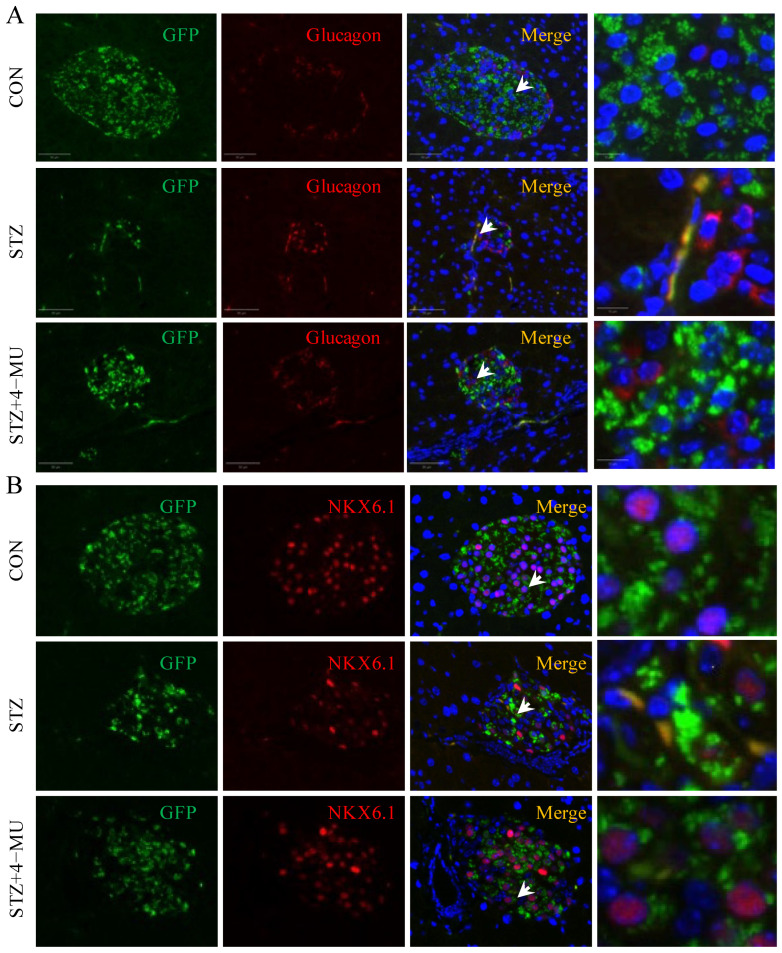
Immunofluorescent analysis of β cell dedifferentiation in the pancreatic tissues of β cell lineage-tracing diabetic mice treated with 4-MU for 3 weeks. (**A**) Representative image of an islet immunostained with eGFP (β cell lineage-tracing marker) and glucagon. Scale bars, 50 μm. (**B**) Representative image of an islet immunostained with eGFP (β cell lineage-tracing marker) and NKX6.1. Arrows indicate GFP/Glucagon^+^ cells or NKX6.1 cells with nuclear translocation in GFP cells. Scale bars, 50 μm. The enlarged image of cells is in the small box. Scale bars, 10 μm.

**Figure 5 biomedicines-12-02790-f005:**
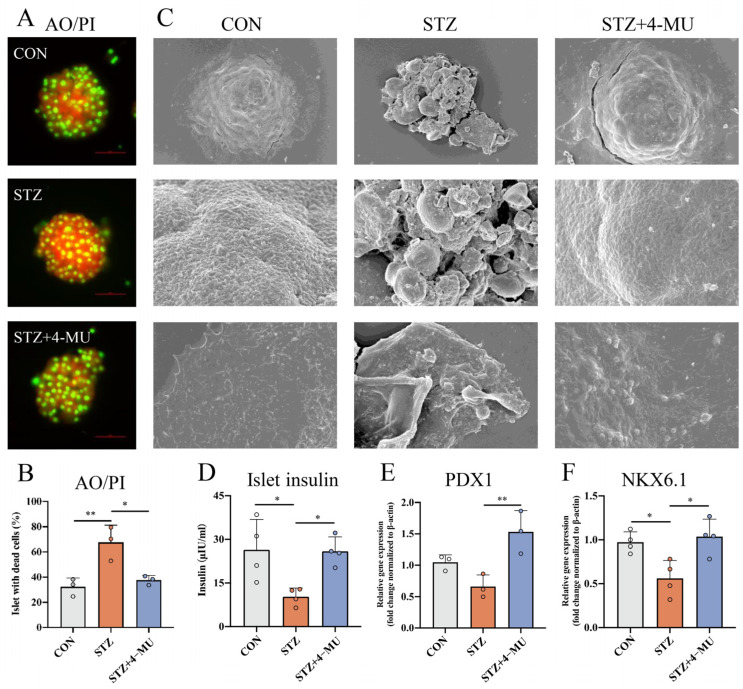
Effect of 4-MU on islet injury induced by STZ in vitro. (**A**) Representative images of AO/PI staining of primary islet, green fluorescence indicates living cells and red fluorescence indicates dead cells. Scale bars, 50 μm. (**B**) Representative images of scanning electron microscope of islets in different groups. Scale bars, 50 μm, 10 μm. (**C**) GSIS experiments in different groups in vitro. (**D**–**F**) Detection of mRNA level of β cells markers. Statistics were determined by two-way analysis of variance (ANOVA) with Tukey’s multiple comparisons test. * *p* < 0.05, ** *p* < 0.01 vs. the STZ.

## Data Availability

All the data, associated protocols and materials for this study are available within the paper.

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
