# Peer review of "Beneficial Actions of 4-Methylumbelliferone in Type 1 Diabetes by Promoting β Cell Renewal and Inhibiting Dedifferentiation"

_biomedicines, 2024, doi:10.3390/biomedicines12122790_

Round 1
Reviewer 1 Report
Comments and Suggestions for Authors
The authors have obtained interesting data, but the following comments
are raised to improve the manuscript:
1. A dose of 50 mg/kg body weight is not low. Please change to
"continuous dose injection, instead of low-dose injection." in the abstract.
2. Please clarify the mutant mice, Ins2Cre/+/Rosa26-eGFP C57BL/6 mice,
in the introduction.
3. A reference to know the assay method of measurement of blood glucose
levels. Also, a reference to the original method of RNA isolation,
reverse transcription, and quantitative PCR.
4. Please add a reference to the Liberase digestion method.
Author Response
Manuscript Number: biomedicines-3305755
Manuscript Title: Beneficial actions of 4-methylumbelliferone in type 1 diabetes by promoting β cells renewal and inhibiting dedifferentiation
We sincerely appreciate the time and efforts that the editor and reviewers spent on reviewing our manuscript. We thank the reviewers for their comments of our work and suggestions to improve the manuscript. We have addressed all of these in the revised manuscript, which have significantly improve the paper quality. For convenience, we have marked the reviewers’ comments in black, our point-to-point response in blue, and corresponding changes in the manuscript in red.
Comment 1: A dose of 50 mg/kg body weight is not low. Please change to "continuous dose injection, instead of low-dose injection." in the abstract.
Response 1: Thank you for your rigorous thinking. According to your suggestions, we have changed “low-dose injection” to “continuous dose injection” in revised manuscript (line 13).
Type 1 diabetes (T1D) model was induced by continuous dose injection of streptozotocin (STZ), and mice were treated with 4-MU for 3 weeks.
Comment 2: Please clarify the mutant mice, Ins2Cre/+/Rosa26-eGFP C57BL/6 mice, in the introduction.
Response 2: Mice were bred in-house by crossing homozygous Rosa26-eGFP mice and Ins2Cre/+ mice (both models on C57BL/6 background) derived from breeding pairs purchased from Cyagen Biosciences. Presence of the Cre and Rosa26-eGFP transgene was confirmed by PCR genotyping. Ins2Cre/+/Rosa26-eGFP C57BL/6 was constructed as a classical islet β cell lineage tracer mouse, which can be used to systematically track insulin-expressing cells. In this study, we used it to monitor the effect of 4-MU on islet b cell dedifferentiation. We have added a description in the introduction of the revised manuscript (line 71-72).
Ins2Cre/+/Rosa26-eGFP C57BL/6 was constructed as a classical islet β cell lineage tracer mouse, which can be used to systematically track insulin-expressing cells.
Comment 3: A reference to know the assay method of measurement of blood glucose levels. Also, a reference to the original method of RNA isolation, reverse transcription, and quantitative PCR.
Response 3: We thank the reviewer for pointing this out. Following the reviewer's suggestion, we have added the reference about the assay method of measurement of blood glucose levels (line 120) and the original method of RNA isolation, reverse transcription, and quantitative PCR (line 194) in the revised manuscript.
Comment 4: Please add a reference to the Liberase digestion method.
Response 4: Thank you for your thoughtful comments. We have added the reference in revised manuscript (line 152).
Thank you for all the valuable and helpful comments and suggestions. We hope that our revised manuscript is now suitable for publication in Biomedicines.
Once again, thank you very much for your comments and suggestions.
Reviewer 2 Report
Comments and Suggestions for Authors
Date: 17-11-2024
Manuscript: Biomedicines 3305755
The authors submitted an article entitled as “Beneficial actions of 4-methylumbelliferone in type 1 diabetes 2 by promoting β cells renewal and inhibiting dedifferentiation” and addressed the impact of 4-methylumbelliferone in type 1 diabetes 2 by promoting β cells renewal and inhibiting dedifferentiation. The work is interesting and well written with informative content. However, I suggest to consider my comments before publication if accepted.
Comment 1. Dear authors I suggest to compare your finding with already published report entitled as “Inhibition of hyaluronan synthesis prevents β-cell loss in obesity-associated type 2 diabetes, https://doi.org/10.1016/j.matbio.2023.09.003”. Is the hypothesis based on different mechanistic perspective or same. Please elaborate.
Comment 2. The sentence “Treatment with 4- 19MU significantly decreased blood glucose and increased plasma insulin levels in STZ-induced diabetes” must be quantitative explanation. Moreover, result and discussion in the abstract is shortly written. Please explain as comparative assessment against the control.
Comment 3. Introduction section is needed to expand with recent finding and challenges. What was the basis for the study, challenges faced during the study, and the clinical expected benefit of the explored compound? I did not find an exhaustive review for the title selection and the objectives.
Comment 4. Abbreviation must be expanded. What is “50–60% r.h”? Did the authors follow ARRIVE guideline? If yes, please explain the blood sampling procedure, volume, time interval, mode of animal sacrifice, and the maximum daily blood sampling.
Comment 5. What was the basis for dose selection (STZ (50 mg/kg)? The source and model of the instruments (like immunofluorescence) must be provided in the manuscript.
Comment 6. In the sentence “Blood samples were obtained from the tail vein of mice”, I recommend to write the sample volume and frequency. What was the sampling schedule followed?
Comment 7. Section 2.5 is shortly written. I suggest to expand considering each step of histopathological assessment. I did not find staining and the processing of the sample slicing.
Comment 8. Please separate unit from its value like “4 %” not “4%”. The “room temperature” must be defined.
Comment 9. In the sentence “Mouse pancreatic islets were isolated using the Liberase digestion method. Briefly, 131mice were anesthetized by intraperitoneal injection of 10% chloral hydrate and the pan- 132creas was perfused with 3 mL collagenase” the authors mentioned chloral hydrate to anaesthetize mice. I did not find chloral hydrate as per new ethical guideline of ARRIVE guidelines. Please double check it.
Comment 10. In the sentence “Serum samples were collected by centrifugation of whole blood at 1000×g for 10 min”, there is no need to centrifuge the collected blood. It needs to collect in a tube without anticoagulant. Moreover, collection of the blood from mice tail is hardly limited. How did the authors collect the blood enough for serum?.
Author Response
Manuscript Number: biomedicines-3305755
Manuscript Title: Beneficial actions of 4-methylumbelliferone in type 1 diabetes by promoting β cells renewal and inhibiting dedifferentiation
We sincerely appreciate the time and efforts that the editor and reviewers spent on reviewing our manuscript. We thank the reviewers for their comments of our work and suggestions to improve the manuscript. We have addressed all of these in the revised manuscript, which have significantly improve the paper quality. For convenience, we have marked the reviewers’ comments in black, our point-to-point response in blue, and corresponding changes in the manuscript in red.
Comment 1: Dear authors I suggest to compare your finding with already published report entitled as “Inhibition of hyaluronan synthesis prevents β-cell loss in obesity-associated type 2 diabetes, https://doi.org/10.1016/j.matbio.2023.09.003”. Is the hypothesis based on different mechanistic perspective or same. Please elaborate.
Response 1: Thanks a lot for the constructive and careful suggestion. Our study differs from the mentioned report in the following three aspects, including model, results and mechanism. Firstly, the mentioned report mainly focused on a mouse model of T2D induced by HFD and STZ. Secondly, the report pointed out the role of 4-MU as an inhibitor of the inflammatory extracellular matrix polymer hyaluronan (HA) synthase ameliorating β cell loss. Finally, it was pointed out to indicate that 4-MU may protect against β-cell injury by increasing the expression of the apoptosis inhibitor survivin. In contrast, first, our study focused on 4-MU on STZ-induced T1D model. Second, the mechanism of action is closely related to the fact that 4-MU promotes β-cell self-replication, inhibits dedifferentiation and immune progression. To clarify this, we have added a discussion in the Discussion section of the revised manuscript (lines 362-371).
Previous studies have demonstrated that 4-MU can ameliorate T2D, suggesting that its use as a hyaluronic acid inhibitor not only alleviates HA deposition in T2D islets, but also may protect β cells from injury by increasing the expression of survivin, an inhibitor of apoptosis. Our latest study further showed that 4-MU also maintains the functional state of β cells by inhibiting their dedifferentiation and promoting β cells regeneration. This means that 4-MU has a dual role in not only protecting the integrity of β cells, but also promoting insulin secretion to help cope with the degradation of β cells function due to immune destruction and chronic stress. 4-MU is expected to be a key factor in effective long-term glycemic control through maintaining the functional properties of β cells.
Comment 2: The sentence “Treatment with 4-MU significantly decreased blood glucose and increased plasma insulin levels in STZ-induced diabetes” must be quantitative explanation. Moreover, result and discussion in the abstract is shortly written. Please explain as comparative assessment against the control.
Response 2: Thank you for your careful reading and thinking. We apologize for not including a comparison and quantitative description of the results with the con group in the abstract. We have made some changes in the abstract part of the revised manuscript (line 21-24).
The plasma insulin level in the STZ group was 7.211 ± 2.602 ng/mL, which was significantly lower than the baseline level (26.94 ± 4.300 ng/mL, p < 0.001). In contrast, the plasma insulin level in the STZ + 4-MU group was 22.29 ± 7.791 ng/mL, which was significantly higher than the STZ group (p < 0.05).
Comment 3: Introduction section is needed to expand with recent finding and challenges. What was the basis for the study, challenges faced during the study, and the clinical expected benefit of the explored compound? I did not find an exhaustive review for the title selection and the objectives.
Response 3: Thanks for the thoughtful comments. This study is an important expansion of the pharmacological effects of 4-MU. It has been previously reported that 4-MU can improve autoimmune diseases, metabolic diseases and even cancers, and also increase brown fat capacity and improve the development of hyperglycemia and obesity. Therefore, we hypothesize that 4-MU can improve the T1D, but the mechanism is not known. In the later research, we mainly focused on the role of 4-MU in the identity transformation of islet β cells in T1D, its accurate location and tracking of β cells is a big challenge for our research. In this work, we hope to find more effective and economical treatment methods and drugs for T1D, providing more choices for clinical improvement and treatment of T1D. We have refined this in the revised manuscript (line 50-55, 61-66, 77-78).
Current treatments for T1D focus on controlling blood glucose levels through insulin therapy and lifestyle changes. However, these approaches fail to address the underlying cause of the disease, which is the progressive loss of functional β-cells. Therefore, new strategies aimed at promoting β-cell regeneration and preventing dedifferentiation are essential to achieve a cure or long-term remission of T1D.
4-MU has been reported to have a great impact in autoimmune diseases, metabolic disorders and even cancer, and can also increase brown fat capacity while ameliorating the development of hyperglycemia and obesity, as well as promoting the restoration of immune tolerance during autoimmune insulitis in T1D mice.
Here, the main aim of this study was to investigate the beneficial effects of 4-MU in T1D by promoting β-cell renewal and inhibiting dedifferentiation, which will provide a basis for the development of new therapeutic strategies for the management and potential cure of T1D.
Comment 4: Abbreviation must be expanded. What is “50–60% r.h”? Did the authors follow ARRIVE guideline? If yes, please explain the blood sampling procedure, volume, time interval, mode of animal sacrifice, and the maximum daily blood sampling.
Response 4: Thank you for your careful reading and rigorous thinking. "50–60% r.h." stands for "50–60% relative humidity", we have made the change in the revised manuscript (line 87). In addition, all of our animal experiments were performed according to the ARRIVE guidelines and were approved by the Institutional Animal Care and Use Committee (Approval NO. IACUC-20230955) of Forth Military Medical University (Page 2, line 89). Blood samples were obtained from the tail vein of mice at the same time each week. Briefly, clip 1-2 mm from the tip of the mouse tail, discard the first drop, and use the second drop to test blood glucose levels (Page 3, line 118-120). Eventually the mice were euthanized by inhalation of CO2 followed by cervical dislocation of the animal to ensure a quick and humane death. The maximum volume of blood collected per animal per day was not more than 10% of the total blood volume (approximately 1% of body weight) in accordance with ethical guidelines to ensure animal welfare.
They were maintained under standard conditions (12 h light–dark cycle; 23–25 °C; 50–60% relative humidity).
All experimental operations were approved by the Animal Care and Use Committee of Forth Military Medical University(Approval NO. IACUC-20230955).
Blood samples were obtained from the tail vein of mice at the same time each week. Briefly, clip 1-2 mm from the tip of the mouse tail, discard the first drop, and use the second drop to test blood glucose levels. Non-fasting blood glucose concentration was measured using a glucometer (Roche Diagnostics, Inc.).
Comment 5: What was the basis for dose selection (STZ (50 mg/kg)? The source and model of the instruments (like immunofluorescence) must be provided in the manuscript.
Response 5: Thank you for your careful reading and rigorous thinking. Continuous intraperitoneal injection of streptozotocin (50mg/kg) once a day for 5 days is a common scheme. A large number of studies have shown that STZ in this dosage range can effectively induce the characteristics of type 1 diabetes, such as hyperglycemia[1,2] (line 94-97). In addition, we have added the source and model of an instrument in the revised manuscript (line 142).
Comment 6: In the sentence “Blood samples were obtained from the tail vein of mice”, I recommend to write the sample volume and frequency. What was the sampling schedule followed?
Response 6: Thank you for your careful reading. We have added relevant method descriptions in the revised manuscript (line 118-120, 107-109).
Blood samples were obtained from the tail vein of mice at the same time each week. Briefly, clip 1-2 mm from the tip of the mouse tail, discard the first drop, and use the second drop to test blood glucose levels. Non-fasting blood glucose concentration was measured using a glucometer (Roche Diagnostics, Inc.).
After a continuous 3-week feeding, an intraperitoneal glucose tolerance test (IPGTT) was performed and samples were collected.
Comment 7: Section 2.5 is shortly written. I suggest to expand considering each step of histopathological assessment. I did not find staining and the processing of the sample slicing.
Response 7: Thank you very much for the excellent comments. According to your suggestions, we have added a detailed description of the experimental method of histopathological assessment in the revised manuscript (Page 3, line 131-133).
Excised pancreatic tissues were fixed with 4 % paraformaldehyde overnight, cut into 5-um-thick sections, and stained with 0.1 % hematoxylin for 15 min and 0.5 % eosin for 5 min, both at room temperature, according to standard procedures. The pathological and morphological evaluation was further carried out by microscope.
Comment 8: Please separate unit from its value like “4 %” not “4%”. The “room temperature” must be defined.
Response 8: Thanks a lot for the excellent comments and suggestions. We have made corrections and additions in the revised manuscript (line 130, 137).
Comment 9: In the sentence “Mouse pancreatic islets were isolated using the Liberase digestion method. Briefly, 131mice were anesthetized by intraperitoneal injection of 10% chloral hydrate and the pan- 132creas was perfused with 3 mL collagenase” the authors mentioned chloral hydrate to anaesthetize mice. I did not find chloral hydrate as per new ethical guideline of ARRIVE guidelines. Please double check it.
Response 9: Thank you for your careful reading and rigorous thinking. We are sorry for describing this method incorrectly. After careful examination, we anesthetized the mice with pentobarbital sodium solution, which has been corrected in the revised manuscript (line 153).
Mouse pancreatic islets were isolated using the Liberase digestion method. Briefly, mice were anesthetized by intraperitoneal injection of sodium pentobarbital solution (100mg/kg body weight) and the pancreas was perfused with 3 mL collagenase P (1 mg/mL, Sigma-Aldrich, USA). The islets were handpicked after purification with Histopaque 1077 (Sigma-Aldrich, USA) density gradients. The islets were isolated and washed twice with 5 mL HBSS buffer. These islets were then cultured overnight in RPMI 1640 containing 10% FBS for further experiment.
Comment 10: In the sentence “Serum samples were collected by centrifugation of whole blood at 1000×g for 10 min”, there is no need to centrifuge the collected blood. It needs to collect in a tube without anticoagulant. Moreover, collection of the blood from mice tail is hardly limited. How did the authors collect the blood enough for serum?
Response 10: Thank you for your careful reading and rigorous thinking. We have corrected the relevant experimental details in the revised manuscript (line 172-177).
At the end of the experimental period, the mice were dislocated, and the whole blood (about 2ml) was collected by eyeball removal in a centrifuge tube without anticoagulant. The collected blood samples were allowed to stand at room temperature for 1 hour or overnight at 4°C to allow serum precipitation. Afterwards, the samples were centrifuged at 4°C, 1000×g for 20 minutes and the supernatant was collected for ELISA.
Thank you for all the valuable and helpful comments and suggestions. We hope that our revised manuscript is now suitable for publication in Biomedicines.
Once again, thank you very much for your comments and suggestions.
Reference:
- Saito, M.; Kinoshita, Y.; Satoh, I.; Shinbori, C.; Suzuki, H.; Yamada, M.; Watanabe, T.; Satoh, K. Ability of cyclohexenonic long-chain fatty alcohol to reverse diabetes-induced cystopathy in the rat. Eur Urol 2007, 51, 479-487; discussion 487-478, doi:10.1016/j.eururo.2006.06.024.
- Arunachalam, K.; Parimelazhagan, T. Antidiabetic activity of Ficus amplissima Smith. bark extract in streptozotocin induced diabetic rats. J Ethnopharmacol 2013, 147, 302-310, doi:10.1016/j.jep.2013.03.004.
Reviewer 3 Report
Comments and Suggestions for Authors
The manuscript presented is properly written and constructed; nonetheless, the following comments should be addressed.
1. According to the literature, the concentrations and durations of STZ vary between studies employing invitro and in vivo models, therefore how did the researchers select their STZ concentrations and durations? The rationale should be given in the manuscript.
2. How has the concentration of 4-MU been determined? The justification should be in the manuscript.
3. the researcher began administering 4-MU after STZ treatment, hypothesizing that 4-MU may reverse the effect of STZ treatments, but throughout their manuscript, they were mentioning a preventive effect of 4-MU against STZ treatment; if the effect is preventive, then 4-MU should have been administered either before or at the same time with STZ, any justifications.
4. in line 179, i think i it is fasting glucose not Non-fasting glucose
Author Response
Manuscript Number: biomedicines-3305755
Manuscript Title: Beneficial actions of 4-methylumbelliferone in type 1 diabetes by promoting β cells renewal and inhibiting dedifferentiation
We sincerely appreciate the time and efforts that the editor and reviewers spent on reviewing our manuscript. We thank the reviewers for their comments of our work and suggestions to improve the manuscript. We have addressed all of these in the revised manuscript, which have significantly improve the paper quality. For convenience, we have marked the reviewers’ comments in black, our point-to-point response in blue, and corresponding changes in the manuscript in red.
Comment 1: According to the literature, the concentrations and durations of STZ vary between studies employing invitro and in vivo models, therefore how did the researchers select their STZ concentrations and durations? The rationale should be given in the manuscript.
Response 1: Thank you for your valuable advice. In the study of in vivo and in vitro models, we have determined the concentration and duration of STZ by referring to high-level articles. The basic principle has been added in the revised manuscript (line 94-98; line 161-163).
In the in vivo model, continuous intraperitoneal injection of streptozotocin (50mg/kg) once a day for 5 days is a common scheme. A large number of studies have shown that STZ in this dosage range can effectively induce the characteristics of type 1 diabetes, such as hyperglycemia[1,2].
In the in vitro model, isolated islets were cultured in 48-well plates for another 24 hours, and about 10 islets per well were supplemented with STZ (1 mM) and 4-MU (0.3 mM) in RPMI 1640. The time selection of STZ dose also refers to related literature[3,4].
Comment 2: How has the concentration of 4-MU been determined? The justification should be in the manuscript.
Response 2: Thank you for your careful reading and excellent suggestions. The intervention dose of 4-MU mainly referred to the relevant references on the regulation of metabolic diseases by 4-MU[5,6] (line 106-107).
Reference (doi:10.1016/j.matbio.2023.09.003) suggests that T2D mice are fed with 250mg/mouse 4-MU every day, which increases the expression of apoptosis inhibitor survivin on β cells, thus improving T2D.
Reference (doi:10.1172/JCI79271) suggests that by feeding autoimmune diabetic mice with 250mg/mouse 4-MU every day, the immune tolerance during autoimmune inflammation can be restored by inhibiting the deposition of endogenous acid.
Comment 3: the researcher began administering 4-MU after STZ treatment, hypothesizing that 4-MU may reverse the effect of STZ treatments, but throughout their manuscript, they were mentioning a preventive effect of 4-MU against STZ treatment; if the effect is preventive, then 4-MU should have been administered either before or at the same time with STZ, any justifications.
Response 3: Thank you for your careful reading and thoughts. We apologize that the description in our manuscript was confusing for you. In fact, according to the design of this study, the use of 4-MU was initiated only after STZ treatment (line 104-106), which demonstrated that 4-MU promoted the renewal and inhibited dedifferentiation of islet β cells in mice with STZ-induced T1D. In the revised manuscript, we have unified the therapeutic effect of 4-MU. However, our study could not show the preventive effect of 4-MU, which will be the next step of our research to explore.
Mice were randomly divided into three experimental groups: (I) CON group, (II) STZ group and (III) STZ + 4-MU group. Mice in the STZ group and the STZ + 4-MU group were intraperitoneally (i.p.) injected with 0.2 mL STZ (50 mg/kg, Sigma-Aldrich, St. Louis, MO, USA) for 5 consecutive days, and the mice in the CON group were intraperitoneally injected with 0.2 mL saline. After T1D mice were successfully constructed, the control and STZ groups were fed with regular chow, and the STZ + 4-MU group were fed with regular chow supplemented with 4-MU (250 mg/mouse/day, Alfa Aesar, Ward Hill, MA).
Comment 4: in line 179, i think i it is fasting glucose not Non-fasting glucose.
Response 4: Thank you for your careful reading. In our study, we detected the random blood glucose level of mice at several time points, and the experimental method is shown in 2.3 (line 117-122), so it is expressed as non-fasting blood glucose.
Thank you for all the valuable and helpful comments and suggestions. We hope that our revised manuscript is now suitable for publication in Biomedicines.
Once again, thank you very much for your comments and suggestions.
Reference:
- Saito, M.; Kinoshita, Y.; Satoh, I.; Shinbori, C.; Suzuki, H.; Yamada, M.; Watanabe, T.; Satoh, K. Ability of cyclohexenonic long-chain fatty alcohol to reverse diabetes-induced cystopathy in the rat. Eur Urol 2007, 51, 479-487; discussion 487-478, doi:10.1016/j.eururo.2006.06.024.
- Arunachalam, K.; Parimelazhagan, T. Antidiabetic activity of Ficus amplissima Smith. bark extract in streptozotocin induced diabetic rats. J Ethnopharmacol 2013, 147, 302-310, doi:10.1016/j.jep.2013.03.004.
- Lee, Y.S.; Song, G.J.; Jun, H.S. Betacellulin-Induced alpha-Cell Proliferation Is Mediated by ErbB3 and ErbB4, and May Contribute to beta-Cell Regeneration. Front Cell Dev Biol 2020, 8, 605110, doi:10.3389/fcell.2020.605110.
- Zhang, J.; Tan, S.B.; Guo, Z.G. CD47 decline in pancreatic islet cells promotes macrophage-mediated phagocytosis in type I diabetes. World J Diabetes 2020, 11, 239-251, doi:10.4239/wjd.v11.i6.239.
- Nagy, N.; Kaber, G.; Sunkari, V.G.; Marshall, P.L.; Hargil, A.; Kuipers, H.F.; Ishak, H.D.; Bogdani, M.; Hull, R.L.; Grandoch, M.; et al. Inhibition of hyaluronan synthesis prevents β-cell loss in obesity-associated type 2 diabetes. Matrix Biology 2023, 123, 34-47, doi:10.1016/j.matbio.2023.09.003.
- Nagy, N.; Kaber, G.; Johnson, P.Y.; Gebe, J.A.; Preisinger, A.; Falk, B.A.; Sunkari, V.G.; Gooden, M.D.; Vernon, R.B.; Bogdani, M.; et al. Inhibition of hyaluronan synthesis restores immune tolerance during autoimmune insulitis. J Clin Invest 2015, 125, 3928-3940, doi:10.1172/JCI79271.
Reviewer 4 Report
Comments and Suggestions for Authors
In this manuscript, the authors used a streptozotocin (STZ) induced type 1 diabetes (T1D) model to investigate the potential beneficial effects of 4-methylumbelliferone (MU). The effects were evaluated both in vivo and in vitro, demonstrating that 4-MU significantly reduces STZ induced beta cell destruction by immune cells. In addition, 4-MU was shown to inhibit beta cell dedifferentiation while maintaining their insulin secretion. Here are some comments.
1. In Fig. 3A and 3B, the authors used CD3 and Ki67 staining to evaluate immune cell infiltration and beta cell proliferation within the islets. It is expected that these protein expressions have significant differences. However, these two protein expressions were also low in CON and STZ+4-MU. Flow cytometry may provide a more precise quantification for these markers.
2. In Fig. 4B, the authors highlighted that “NKX6.1 appeared in the cytoplasm of beta cells”. What mechanisms could explain this observation? Could this represent the underlying mechanism of 4-MU’s therapeutic effects?
3. In Fig. 5A, It’s hard to evaluate the difference of live and dead cells across different groups. Techniques such as flow cytometry or immunoblotting for death markers could strengthen the conclusion.
4. Comparison of Fig. 4B and Fig. 5E. In Fig. 4B, NKX6.1 expression levels of STZ and STZ+-4MU did not show an obvious difference, but the subcellular location differed. NKX6.1 level in STZ+4-MU appeared lower than in the CON group. In contrast, Fig. 5E shows NKX6.1 RNA level in STZ+4-MU seems higher than in the CON group. The authors should confirm whether additional regulatory mechanisms underlie this apparent imbalance between RNA and protein levels.
Author Response
Manuscript Number: biomedicines-3305755
Manuscript Title: Beneficial actions of 4-methylumbelliferone in type 1 diabetes by promoting β cells renewal and inhibiting dedifferentiation
We sincerely appreciate the time and efforts that the editor and reviewers spent on reviewing our manuscript. We thank the reviewers for their comments of our work and suggestions to improve the manuscript. We have addressed all of these in the revised manuscript, which have significantly improve the paper quality. For convenience, we have marked the reviewers’ comments in black, our point-to-point response in blue, and corresponding changes in the manuscript in red.
Comment 1: In Fig. 3A and 3B, the authors used CD3 and Ki67 staining to evaluate immune cell infiltration and beta cell proliferation within the islets. It is expected that these protein expressions have significant differences. However, these two protein expressions were also low in CON and STZ+4-MU. Flow cytometry may provide a more precise quantification for these markers.
Response 1: Thanks a lot for the excellent comments and suggestions. We are very sorry that we did not provide more precise quantification of immune cell infiltration and β-cell proliferation in the islet by flow cytometry.
We chose to use other methods such as HE staining because these methods have been widely used in the detection of islet immune infiltration and cell proliferation, and can effectively evaluate the integrity and immune infiltration of islets (as shown in Figure 1-F). And we drew on multiple references, all suggesting that CD3 staining of pancreatic sections can indicate inflammatory infiltration of pancreatic islets[1-3].
In addition, Ki67 staining has been used alone in several high-level studies to evaluate cell proliferation[4-6]. Although flow cytometry is a commonly used and widely accepted detection method, which can provide more accurate quantitative data, it needs to extract islets, digest them into single cells, sort β cells, and then mark proliferation markers, which is challenging for our experiment. We plan to use flow cytometry for more comprehensive verification in the follow-up study to further support our conclusion.
Comment 2: In Fig. 4B, the authors highlighted that “NKX6.1 appeared in the cytoplasm of beta cells”. What mechanisms could explain this observation? Could this represent the underlying mechanism of 4-MU’s therapeutic effects?
Response 2: Thank you for your insightful thoughts and comments. NKX6.1 is a key transcription factor that is essential for maintaining β-cell identity[7]. It helps β-cells resist stress and inflammation by regulating the expression of several genes, thereby maintaining their specific insulin secretory function and preventing dedifferentiation[8]. ERS, oxidative stress, and inflammatory responses all induce NKX6.1 nuclear translocation[9]. When NKX6.1 translocate from the nucleus to the cytoplasm, it fails to effectively regulate the expression of insulin genes and other key genes, leading to the loss of its specific function and dedifferentiation of β-cells[10]. In the T1D model, NKX6.1 appeared in the cytoplasm instead of the nucleus, suggesting that it might be related to the dedifferentiation of β-cells[11]. Our study suggests that 4-MU can inhibit the dedifferentiation of β-cells by suppressing the nuclear translocation abnormality of NKX6.1, and the potential mechanism of its therapeutic effect may be closely related to ERS, oxidative stress and inflammatory response, which will be explored and verified in depth in the subsequent studies.
Comment 3: In Fig. 5A, it’s hard to evaluate the difference of live and dead cells across different groups. Techniques such as flow cytometry or immunoblotting for death markers could strengthen the conclusion.
Response 3: We thank the reviewer for pointing this out. As shown in Figure 5-1, the islet activity in vitro was detected by AO/PI staining, in which green fluorescence indicates living cells and red fluorescence indicates dead cells. The results suggest that 4-MU can significantly alleviate the islet damage induced by STZ and restore islet activity. Although the death markers detected by flow cytometry and western blot can strengthen this conclusion, our research mainly focuses on the effect of 4-MU on the dedifferentiation and function of islet β cells, so we think this method can support the conclusion, and AO/PI has also been used alone to evaluate cell viability in some high-level publications[12-14]. In subsequent studies, we will take flow cytometry or immunoblotting to support the conclusion more comprehensively.
Comment 4: Comparison of Fig. 4B and Fig. 5E. In Fig. 4B, NKX6.1 expression levels of STZ and STZ+-4MU did not show an obvious difference, but the subcellular location differed. NKX6.1 level in STZ+4-MU appeared lower than in the CON group. In contrast, Fig. 5E shows NKX6.1 RNA level in STZ+4-MU seems higher than in the CON group. The authors should confirm whether additional regulatory mechanisms underlie this apparent imbalance between RNA and protein levels.
Response 4: Thank you for your careful reading and rigorous thinking. Figures 4-B and 5-E support that 4-MU can effectively inhibit the dedifferentiation of β cells by regulating the expression of NKX6.1 and inhibiting its abnormal nuclear metastasis.
Figure 4-B shows the immunofluorescence results of co-localization of insulin and NKX6.1 in pancreatic tissues of the T1D mouse model, and we determined the dedifferentiation of β-cell by detecting the nucleoplasm distribution of NKX6.1 in insulin+ cells. As shown in Figure 4-B, there was no significant difference in the proportion of insulin+ cells between the CON and 4-MU groups, but the expression of insulin+ cells was significantly decreased in the STZ group, and the NKX6.1 of insulin+ cells in the STZ group was more distributed in the cytoplasm, but the NKX6.1 of the 4-MU and CON group was uniformly distributed in the nucleus. These results suggest that STZ induces beta-cell dedifferentiation, which can be ameliorated by 4-MU.
Figure 5-E shows the gene expression level of NKX6.1 in primary islets after in vitro intervention., The results suggest that STZ can reduce the expression of NKX6.1, while 4-MU can restore the expression of NKX6.1, thus maintaining the identity and function of β-cell.
In addition, the differences between RNA and protein levels may due to multiple regulatory mechanisms, including post-transcriptional modifications, protein stability, and changes in subcellular localization. We plan to further explore these mechanisms in subsequent studies, especially through mass spectrometry and immunoprecipitation, to verify the inconsistency between RNA and protein expression and to reveal the mechanism of action of 4-MU in greater depth.
Thank you for all the valuable and helpful comments and suggestions. We hope that our revised manuscript is now suitable for publication in Biomedicines.
Once again, thank you very much for your comments and suggestions.
Reference:
- Apaolaza, P.S.; Balcacean, D.; Zapardiel-Gonzalo, J.; Rodriguez-Calvo, T. The extent and magnitude of islet T cell infiltration as powerful tools to define the progression to type 1 diabetes. Diabetologia 2023, 66, 1129-1141, doi:10.1007/s00125-023-05888-6.
- Yin, N.; Zhang, N.; Xu, J.; Shi, Q.; Ding, Y.; Bromberg, J.S. Targeting lymphangiogenesis after islet transplantation prolongs islet allograft survival. Transplantation 2011, 92, 25-30, doi:10.1097/TP.0b013e31821d2661.
- Lundberg, M.; Seiron, P.; Ingvast, S.; Korsgren, O.; Skog, O. Insulitis in human diabetes: a histological evaluation of donor pancreases. Diabetologia 2017, 60, 346-353, doi:10.1007/s00125-016-4140-z.
- Mao, D.; Cao, H.; Shi, M.; Wang, C.C.; Kwong, J.; Li, J.J.X.; Hou, Y.; Ming, X.; Lee, H.M.; Tian, X.Y.; et al. Increased co-expression of PSMA2 and GLP-1 receptor in cervical cancer models in type 2 diabetes attenuated by Exendin-4: A translational case-control study. EBioMedicine 2021, 65, 103242, doi:10.1016/j.ebiom.2021.103242.
- Nel, I.; Beaudoin, L.; Gouda, Z.; Rousseau, C.; Soulard, P.; Rouland, M.; Bertrand, L.; Boitard, C.; Larger, E.; Lehuen, A. MAIT cell alterations in adults with recent-onset and long-term type 1 diabetes. Diabetologia 2021, 64, 2306-2321, doi:10.1007/s00125-021-05527-y.
- Park, Y.G.; Lee, J.Y.; Kim, C.; Park, Y.H. Early Microglial Changes Associated with Diabetic Retinopathy in Rats with Streptozotocin-Induced Diabetes. J Diabetes Res 2021, 2021, 4920937, doi:10.1155/2021/4920937.
- Aigha, II; Abdelalim, E.M. NKX6.1 transcription factor: a crucial regulator of pancreatic beta cell development, identity, and proliferation. Stem Cell Res Ther 2020, 11, 459, doi:10.1186/s13287-020-01977-0.
- Brusco, N.; Sebastiani, G.; Di Giuseppe, G.; Licata, G.; Grieco, G.E.; Fignani, D.; Nigi, L.; Formichi, C.; Aiello, E.; Auddino, S.; et al. Intra-islet insulin synthesis defects are associated with endoplasmic reticulum stress and loss of beta cell identity in human diabetes. Diabetologia2023, 66, 354-366, doi:10.1007/s00125-022-05814-2.
- Dos Santos, C.; Cambraia, A.; Shrestha, S.; Cutler, M.; Cottam, M.; Perkins, G.; Lev-Ram, V.; Roy, B.; Acree, C.; Kim, K.Y.; et al. Calorie restriction increases insulin sensitivity to promote beta cell homeostasis and longevity in mice. Nat Commun 2024, 15, 9063, doi:10.1038/s41467-024-53127-2.
- Liu, T.; Sun, P.; Zou, J.; Wang, L.; Wang, G.; Liu, N.; Liu, Y.; Ding, X.; Zhang, B.; Liang, R.; et al. Increased frequency of beta cells with abnormal NKX6.1 expression in type 2 diabetes but not in subjects with higher risk for type 2 diabetes. BMC Endocr Disord 2021, 21, 47, doi:10.1186/s12902-021-00708-7.
- Isildar, B.; Ozkan, S.; Ercin, M.; Gezginci-Oktayoglu, S.; Oncul, M.; Koyuturk, M. 2D and 3D cultured human umbilical cord-derived mesenchymal stem cell-conditioned medium has a dual effect in type 1 diabetes model in rats: immunomodulation and beta-cell regeneration. Inflamm Regen 2022, 42, 55, doi:10.1186/s41232-022-00241-7.
- Xu, M.; Wen, Y.; Liu, Y.; Tan, X.; Chen, X.; Zhu, X.; Wei, C.; Chen, L.; Wang, Z.; Liu, J. Hollow mesoporous ruthenium nanoparticles conjugated bispecific antibody for targeted anti-colorectal cancer response of combination therapy. Nanoscale 2019, 11, 9661-9678, doi:10.1039/c9nr01904a.
- Cheng, P.; Jian, Q.; Fu, Z.; Deng, R.; Ma, Y. Inhibition of DAI refrains dendritic cells from maturation and prolongs murine islet and skin allograft survival. Front Immunol 2023, 14, 1182851, doi:10.3389/fimmu.2023.1182851.
- Wang, J.; Wang, J.; Wang, Y.; Ma, R.; Zhang, S.; Zheng, J.; Xue, W.; Ding, X. Bone Marrow Mesenchymal Stem Cells-Derived miR-21-5p Protects Grafted Islets Against Apoptosis by Targeting PDCD4. Stem Cells 2023, 41, 169-183, doi:10.1093/stmcls/sxac085.
Round 2
Reviewer 2 Report
Comments and Suggestions for Authors
recommended for publication
Author Response
Thank you for your work and for the recognition of our study.
Reviewer 4 Report
Comments and Suggestions for Authors
Thanks to the authors for answering my comments in the revised manuscript. In Point 1, I agree with the authors that Ki-67 can be used to evaluate cell proliferation. However, in their figure, there is only a difference of one or two cells, which lacks general convincing evidence. In Point 3, the assessment of live cells using AO/PI staining is good. However, relying on visual inspection alone makes it difficult to demonstrate a significant reduction in islet damage. That’s why I recommended the authors to quantify the staining. In addition, I am confused about the explanation of the inconsistency of figure 4B and 5E. The authors emphasized figure 4B is pancreatic tissues from the mouse model, while figure 5E is in vitro intervention. My understanding was that both figures showed primary islets from mouse experiments but used different analyses. If this interpretation is incorrect, please clarify the conditions in the Results section.
Author Response
Dec 5, 2024
Manuscript Number: biomedicines-3305755
Manuscript Title: Beneficial actions of 4-methylumbelliferone in type 1 diabetes by promoting β cells renewal and inhibiting dedifferentiation
We sincerely appreciate the time and efforts that the reviewer spent on reviewing our manuscript. We thank the reviewers for their comments of our work and suggestions to improve the manuscript. We have addressed all of these in the revised manuscript, which have significantly improved the paper quality. For convenience, we have marked the reviewers’ comments in black, our point-to-point response in blue, and corresponding changes in the manuscript in red.
Reviewer #3: Thanks to the authors for answering my comments in the revised manuscript.
Comment 1: In Point 1, I agree with the authors that Ki-67 can be used to evaluate cell proliferation. However, in their figure, there is only a difference of one or two cells, which lacks general convincing evidence.
Response 1: Thank you for your valuable comments and suggestions. We also noticed this phenomenon, but we found that many groups of data all showed the same result. Moreover, relevant references have reported that the proliferation rate of islets is low (Biomedicines. 2022 Jan 18;10(2):203; PLoS Biol. 2007 Oct 16;5(10):e276), thus presenting differences in only a few cells. In addition, we have reviewed and studied the results and figure of other references, which are similar to our results. Examples include the following three references:
Reference 1: Zhu K, Lai Y, Cao H, etc. Kindlin-2 modulates MafA and β-catenin expression to regulate β-cell function and mass in mice. Nat Commun. 2020. doi: 10.1038/s41467-019-14186-y.
The authors performed dual staining with anti-insulin (green) and Ki-67 antibodies (red) on one-week-old control and mutant male islet. The results indicate that the loss of Kindlin-2 decreases β-cell proliferation.
Reference 2: Ding L, Sun Y, Liang Y, etc. Beta-Cell Tipe1 Orchestrates Insulin Secretion and Cell Proliferation by Promoting Gαs/cAMP Signaling via USP5. Adv Sci (Weinh). 2024. doi: 10.1002/advs.202304940.
The team used Insulin (red) and Ki-67 (green) to measure islet β cell proliferation, demonstrating that Tipe1 promotes β cell proliferation.
Reference 3: Díaz-Catalán D, Alcarraz-Vizán G, Castaño C, etc. BACE2 suppression in mice aggravates the adverse metabolic consequences of an obesogenic diet. Mol Metab. 2021. doi: 10.1016/j.molmet.2021.101251.
The report utilized Insulin (green) and Ki67 (red) staining to assess β-cell proliferation in the islets. The findings demonstrate that long-term high-fat diet (HFD) feeding leads to increased β-cell proliferation in BKO mice.
Comment 2: In Point 3, the assessment of live cells using AO/PI staining is good. However, relying on visual inspection alone makes it difficult to demonstrate a significant reduction in islet damage. That’s why I recommended the authors to quantify the staining.
Response 2: Thank you for your valuable comments and suggestions. We have seriously considered the method of evaluating viability using AO/PI staining and agree with you. In order to improve the objectivity and accuracy of the data, we used ImageJ to quantify the results of AO/PI staining. The percentage of islets containing dead cells was quantified to more accurately reflect the changes in islet viability under different treatments. Based on the results of the above quantitative analysis, we added a new statistical analysis (Figure 5) to the manuscript to support our conclusion. The following are the original data and statistical charts of our quantitative analysis. The results show that 4-MU can significantly reduce the islet damage induced by STZ and restore islet activity in vitro.
Figure 5. (A) Representative images of AO/PI staining. Scale bars, 50 μm. (B) After staining with AO/PI, the percent of islets containing dead cells was quantified (mean ± SEM, n = 3/group).
Comment 3: In addition, I am confused about the explanation of the inconsistency of figure 4B and 5E. The authors emphasized figure 4B is pancreatic tissues from the mouse model, while figure 5E is in vitro intervention. My understanding was that both figures showed primary islets from mouse experiments but used different analyses. If this interpretation is incorrect, please clarify the conditions in the Results section。
Response 3: Thank you for your careful thinking and suggestions. We are sorry that the previous results have confused you. Figure 4-B is an immunofluorescence staining of pancreatic tissue sections, which detects the nuclear translocation of NKX6.1 in mouse islet β cells. Figure 5-E shows the expression level of NKX6.1 in primary islets of mice after STZ intervention in vitro. These results suggest that 4-MU inhibits islet β cell dedifferentiation by promoting NKX6.1 expression and inhibiting its nuclear translocation. As you can understand, both Figure 4-B and Figure 5-E show the relevant results for mouse islets, but the assays were performed using different methods in pancreatic tissue sections and primary islets, respectively.
Thank you for all the valuable and helpful comments and suggestions. We hope that our revised manuscript is now suitable for publication in Biomedicines.
Once again, thank you very much for your comments and suggestions.
Yours,
Bin Gao
Round 3
Reviewer 4 Report
Comments and Suggestions for Authors
The authors have adequately addressed my concerns in the revised manuscript.